# Utilization of organic-residues as potting media: Physico-chemical characteristics and their influence on vegetable production

Tajwar Alam [1,2]*, Muhammad Ikram[2], Arshad Nawaz Chaudhry[2], Chaudhry Muhammad Subhan[3], Khaled D. Alotaibi[4], Zia-UI -Haq[3], Muhammad Sohail Yousaf[5], Hasan Pervej Ahmed[6], Nida Fatima[2], Ghulam Jilani[2], Muhammad Shakir Farooq[2], Mohammad Naseem[2], Muhammad Ishaq[2]*

1 Institute of Hydroponic Agriculture, PMAS-Arid Agriculture University, Rawalpindi, Pakistan, 2 Institute of Soil & Environmental Science, PMAS-Arid Agriculture University, Rawalpindi, Pakistan, 3 Faculty of Agriculture Engineering, PMAS-Arid Agriculture University, Rawalpindi, Pakistan, 4 Department of Soil Science, College of Food and Agriculture Sciences, King Saud University, Riyadh, Saudi Arabia, 5 Yale School of the Environment, New Haven, Connecticut, United States of America, 6 Department of Soil Science, University of Saskatchewan, Saskatoon, Canada

* tajwaralam@uaar.edu.pk (TA); muhammadishaq@uaar.edu.pk (MI)

**Data Availability Statement:** All relevant data are within the paper.

**Funding:** This study was funded by the College of Food and Agriculture Sciences, King Saud

## Abstract

Soilless agriculture is acknowledged worldwide because it uses organic leftovers as a means of supporting intensive and efficient plant production. However, the quality of potting media deteriorates because of lower nutrient content and excessive shrinkage of most organic materials. A current study was undertaken to identify the optimal blend of locally available organic materials with desirable qualities for use as potting media. Therefore, different ingredients, viz., *Pinus roxburghii* needles, sugarcane bagasse, and farmyard manure were used alone or in combination as potting media to test their suitability by growing spinach as a test crop. Results showed that an increase in *Pinus roxburghii* needles and sugarcane bagasse decreased medium pH and electrical conductivity. Higher pH and electrical conductivity were recorded for the treatments having a higher farmyard manure ratio ($\geq$50%) in combination. Except for pine needles 100%, pH and electrical conductivity were in the recommended range. The growth attributes include, leaves plant$^{-1}$, shoot length, fresh- and dry shoot weight along with plant macronutrients (nitrogen, phosphorous, and potassium) and micronutrients (iron, copper, manganese, and zinc) content were higher in treatment pine needles 50% +farmyard manure 50% followed by pine needles 25%+farmyard manure 50%+sugarcane bagasse 25%. Moreover, the particular treatment of pine needles 50%+farmyard manure 50% exhibited the highest concentrations of macro- (nitrogen, phosphorus, and potassium) as well as micronutrients (iron, copper, manganese, and zinc) in the potting media following the harvest. This study highlights the potential of utilizing agro-industrial litter/waste as a soilless growing medium for spinach production under greenhouse conditions. When employed in appropriate proportions, this approach not only addresses disposal concerns but also proves effective for sustainable cultivation. Further research is needed to investigate the use of these wastes as potting media by mixing various particle-size ingredients.

University Researchers Supporting Project Number (RSPD2024R633). This funding was recieved by KDA.

**Competing interests:** The authors declare they have no known conflict or competing interest.

## 1. Introduction

Growing plants without soil as a rooting medium is called a soilless plant culture [1]. Compared with soil-based cultivation, it is cost-effective, producing higher crop yield and prompter harvest from smaller pieces of land [2]. Moreover, soilless potting media generally contains higher nutrient and water use efficiency [3]. Soilless culture systems for vegetable production are becoming popular worldwide due to several issues, including pests and diseases in the soil, soil quality, and lack of fresh water for irrigation [4]. As a result, during the past 50 years, their significance has grown on a global scale [5]. However, the quality of the potting media has a major impact on the success of substrate culture. When choosing materials for soilless potting media, it's important to consider factors like adequate drainage, high water retention, high ionic exchange capacity, and lack of pathogens, weeds, and pests [6]. Additionally, for optimal natural recycling, the material should be readily available, less expensive, and ideally organic [7].

Numerous studies have focused on finding a low-cost, locally accessible substance like farmyard manure (FM), particularly as a substitute for peat when producing decorative pot plants [8] and vegetables [9]. However, variability in growing substrate can pose significant problems by affecting growth rate and plant nutrition. The use of organic wastes as potting media protects the environment by decreasing ecosystem damage caused by peat or soil extraction [10] and provides economic benefits, as the use of organic residues like FM is cheaper as compared to conventional materials [11].

*Pinus roxburghii* is a widely distributed conifer in the Himalayas covering around 3000 km with 7.64 million hectares and covers around 0.40 million hectares area only in Pakistan [12]. Pine leaves/needles are waste products [13] which contain various biochemicals that are being used as an antiseptic, and liver tonic [14]. Pine needles are a good choice for potting media due to their suitable physical/chemical properties, availability, and lower cost. Additionally, it contains most of the essential nutrients required for plant growth [15, 16]. Pine needles/bark contain 0.3–0.5% nitrogen (N) [17] and a high C: N ratio ranging from 100 to 200 [18], lower EC, and very acidic, freshly ground, with a pH of roughly 3–4 [19]. Therefore, it can be mixed with other materials like FM to enhance its C: N ratio, pH, and EC for a successful growing media. Sugarcane is among the world's major crops [20] and also the leading crop in Pakistan having the largest sugar market in terms of volume and the sixth largest sugarcane producer (around 87 million tonnes) globally [21]. Among the easily available organic residues, SB is one of the important organic wastes. It is a fibrous, pulpy waste that is a byproduct of the sugarcane industry having (47–52%) cellulose, (25–28%) hemicellulose, and (20–21%) lignin. It produces organic acids, that mobilize insoluble phosphorous (P) in the labile form and possess a significant quantity of essential plant nutrient content [22].

Several grades of perlite, pine bark, and peat moss have been evaluated as soilless growing media by [23] and found that coarse-grade perlite produced more commercially viable cucumber fruit compared to medium-grade perlite. Application of composted organic waste and biochar enhanced plant growth attributes and nutrient contents compared to peat media [24]. Various substrates were evaluated by [25] through growing tomatoes and reported that bark, peat, and peat in combination with bark had yields comparable to those of rock wool. Pine bark and sewage sludge blends were used as substitutes for peat in potting media and reported an increase in plant yield with the addition of organic refuse [26]. Imported peat's high cost and environmental concerns have prompted researchers to look for substitute materials to use as growth substrates [27].

Soilless substrates are accepted alternatives and beneficial in fruit and vegetable production. The antioxidant activity of Chinese cabbage and spinach is increased when an organic substrate is added to the root media [28]. Many researchers have observed higher yield, and

increase in sugar, vitamins, and carotenoid content of tomatoes grown in soilless potting media [25]. Organic matter contents, C: N ratio, bulk density, porosity, CEC, pH, EC and the presence of nutrient content are peculiar characteristics of growing substrate [29].

Therefore, objectives of the current study were to evaluate the physico-chemical properties of potting media derived from organic residues, identify how these properties influence vegetable production, and recommend the optimal utilization of these residues (FM, PN, and SB) as potting media. The hypothesis posited that the physico-chemical properties of potting media, obtained from organic residues, play a significant role in influencing the growth parameters of vegetables due to their richness in organic nutrients. The anticipation was that specific combinations of organic residues within the potting media would lead to optimized conditions conducive to enhanced vegetable production.

## 2. Materials and methods

### 2.1. Site description

The study was undertaken in the greenhouse at Pir Mehr Ali Shah-Arid Agriculture University Rawalpindi, during 2020–2021. The site's GPS location was latitude 32˚17'48 N and longitude 72˚21'9 E with 615m elevation from sea level and 1,346.8 mm annual rainfall. No rainfall was recorded during the experiment. Temperature was maintained at 28±2 ˚C while humidity was 65±5%.

### 2.2. Collection of potting media ingredients and their processing

A range of plant-based components was used in varying quantities to prepare the substrate, viz., falling leaves of *Pinus roxburghii* (PN), FM, and SB alone or in combination. *P. roxburghii* needles were collected from hills close to Islamabad, Pakistan (about 508 meters above sea level at 73˚ 02' E longitude and 33˚ 36' N latitude). The SB was collected from a local sugar mill and FM was collected from a nearby dairy farm. The SB and PN were sundried for 2 weeks in shade around 28–30 ˚C followed by oven drying for 24 h at 60 ˚C. The dried potting material was ground in a rotary grinding mill. To separate the potting mixes' proper particle sizes, a 2 mm size mesh is attached. The ground potting media were filled in 10 kg pots. Each pot was filled with an equal amount of material, viz., 8 kg (by volume). The treatment combination was, PN100%, SB100%, PN50%+SB50%, PN50%+FM50%, SB50%+FM50%, and PN25%+FM50%+SB25%. The experimental design was completely randomized (CRD) with three replications.

### 2.3. Pre-analysis of material used

The material used for potting media was analyzed for various traits before the experiment. Chemical properties of FM are pH (7.54), EC (39 mSm$^{-1}$), total N (0.96%), P (0.46%), potassium (K) (0.84%), copper (Cu) (66.2 mg kg$^{-1}$), manganese (Mn) (141 mg kg$^{-1}$) and zing (Zn) (48.31 mg kg$^{-1}$). Pine leaves/needles chemical composition was recorded as, pH (4.89), EC (11 mSm$^{-1}$), total N (0.22%), P (0.12%), K (0.14%), Cu (13.60 mg kg$^{-1}$), Mn (112 mg kg$^{-1}$) and Zn (33.1 mg kg$^{-1}$). The SB had the following characteristics: pH 6.2, EC 0.14 mSm$^{-1}$, C: N (47:1), N (0.15%), P (0.10%), K (0.13%), Cu (11.60 mg kg$^{-1}$), Mn (102 mg kg$^{-1}$) and Zn (26.1 mg kg$^{-1}$).

### 2.4. Chemical analysis

Potting media pH was determined through pH meter using 1:10 ratios of sample water suspension and EC was analyzed by conductivity meter from supernatant liquid obtained from 1:10 ratio sample water suspension [30]. Muffle furnace was used to determine the organic carbon

and samples dry ashing were carried out for 4.5 h at 550 ˚C. Wet digestion was undertaken to measure P, K, Cu, Mn Zn, and Fe contents. Digestate was used to analyze Cu, Mn, Zn, and Fe concentrations through Atomic Absorption Spectroscopy (AAS) at various wavelengths, viz., Cu (324.8 nm), Mn (279.5 nm), and Zn (213.7 nm). Spectrophotometer was used to analyze P by following [31]. Wet-digested filtrate was obtained and dilution was done as per necessity. Before the analysis of samples working standards were run and the graph was plotted. The samples were run on spectrophotometer and readings were noted at 410 nm wavelength. Potassium concentration in the aliquot was noted on flame photometer by following [32]. The determination of total Kjeldahl-N involved taking 1 g of sample (air-dried) and mixing it with 3.5 g of digestate mixture ($K_2SO_4$: Se) in a digestion tube. To this mixture, 10 mL $H_2SO_4$ (conc.) was added, and the solution was heated for 30 minutes in block digester at 420 ˚C. Subsequently, distillation of digested sample was carried out in a conical flask using a distillation unit. The distillate from compost samples was collected in a 25 mL solution of 4% boric acid to which a few drops of bromocresol-green and ethyl red were added. The filtrate was then titrated using $H_2SO_4$ 0.1 N and the total Kjeldahl-N was recorded by following [33].

## 2.5. Plant characteristics

Spinach was harvested and analyzed for various parameters, viz., shoot length, leaves per plant, shoot fresh weight were recorded and subsequently, dried in an oven for 72 h at 65±5 ˚C, and dry weights were noted. For the determination of P, K, Cu, Mn Zn, and Fe contents samples were ground to fine powder, and wet digestion was done. By using the digestate Cu, Mn, and Zn concentrations were analyzed with AAS. Spectrophotometer was used to determine the phosphorus concentration by following [31] while K concentration was measured in the aliquot using flame photometer following Knudsen et al. (1983). Total N was estimated as described by [33].

## 2.6. Statistical analysis

Data collected for different plant characteristics and potting media were analyzed statistically via Statstix 8.1 under a completely randomized design and LSD was employed to compare treatment means at a significance level of 5% [34]. Graphs were drawn with MS Excel 2010.

# 3. Results

## 3.1. Growth attributes and nutrient content of spinach

The highest (24.3 cm) and lowest (6.46 cm) plant height were obtained at the PN50%+FM50% and SB100% media, respectively (Fig 1a). Plant height was greatly boosted when PN and FM were used in combination compared with PN, SB, or FM alone. Also adding PN and SB to FM significantly increased plant height compared to these materials alone or in combination with PN and SB. The plants' heights grown in PN50%+FM50% and PN25%+FM50%+SB25% were significantly higher than those grown in other treatments/combinations. Plant height was also increased significantly when SB was added to FM in comparison to the SB100%, PN50%+-SB50%, and PN100% media, respectively. The highest fresh shoot biomass (24.3 g plant$^{-1}$) was obtained by using PN50%+FM50% as planting media while the lowest shoot fresh biomass (1.91 g plant$^{-1}$) was obtained for SB100% potting media (Fig 1b). Moreover, plants grown in PN50%+FM50% produced the highest shoot dry biomass and SB100% substrate produced the lowest shoot dry biomass (Fig 1c). Adding PN to FM or PN and SB to FM significantly increased both shoot's fresh and dry biomass of spinach plants compared to the SB and PN solely (Fig 1a and 1b). A comparable pattern was noted for leaves per plant (Fig 1d). The

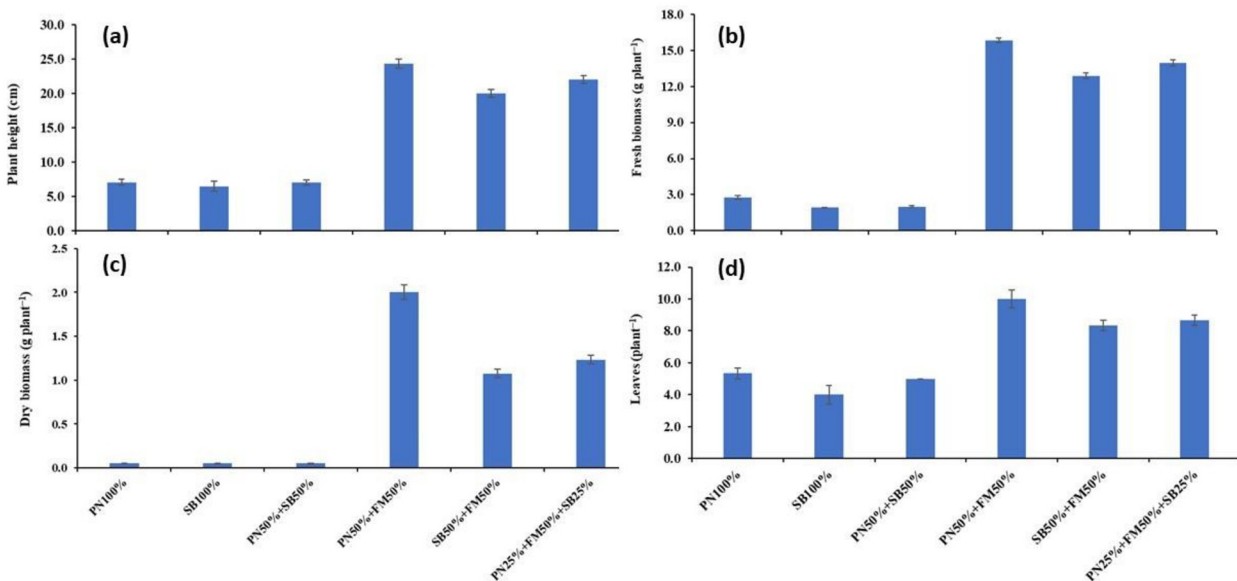

**Fig 1. Effect of substrate composition on plant growth and yield attributes.** Error bars indicate standard errors from mean (n = 3). PN (pine needles); FM (farmyard manure); EM (effective microorganism).

highest and lowest leaves per plant were obtained by using PN50%+FM50% and SB100% as the media, respectively. When PN and SB were added to the FM instead of using PN or SB alone or in combination, the number of leaves per plant increased considerably.

Nutrient contents in plant tissue showed significant responses to potting media. The highest N (3.03%) and K (1.26%) concentration in plant tissue was obtained at FM50%+PN25%+SB25% while the lowest N (2.15%) and K (1.01%) were obtained at the PN010% media (Fig 2a–2c) respectively. In the case of P, the highest P contents in plant tissues were recorded in SB50%+FM50% media while the lowest P contents were noted in plants grown at the PN100% media (Fig 2b). The concoction of SB and PN with FM significantly increased macronutrient contents in plant tissues compared with PN or SB alone.

Micronutrient content in plant tissues is shown in Table 1. The highest Cu and Zn contents were recorded for SB50%+FM50% media while the lowest values were obtained for PN100% (Table 1). In the case of Mn and Fe the highest values were noted for PN50%+FM50% media and the lowest Mn and Fe contents were recorded for SB100% media (Table 1). Using PN or SB alone or in combination produced the lowest nutrient contents in plant tissues however, adding SB and PN to FM significantly increased micronutrient contents.

### 3.2. Physicochemical properties of potting media

Data related to physico-chemical properties has been shown in Table 2. Changes in pH and EC differed significantly among the various treatments. The highest pH (6.80) was recorded in SB50%+FM50% media while the lowest values for pH (6.12) were noted for PN100% media (Table 2). The salinity of potting media expressed as the EC of the saturated media extract varied greatly between potting media. The highest EC (125 dSm$^{-1}$) values were recorded in SB50%+FM50% media while the lowest values for EC (78.7 dSm$^{-1}$) were noted for PN100% media. The PN and SB media showed lower salt content compared to the potting media having FM in combination.

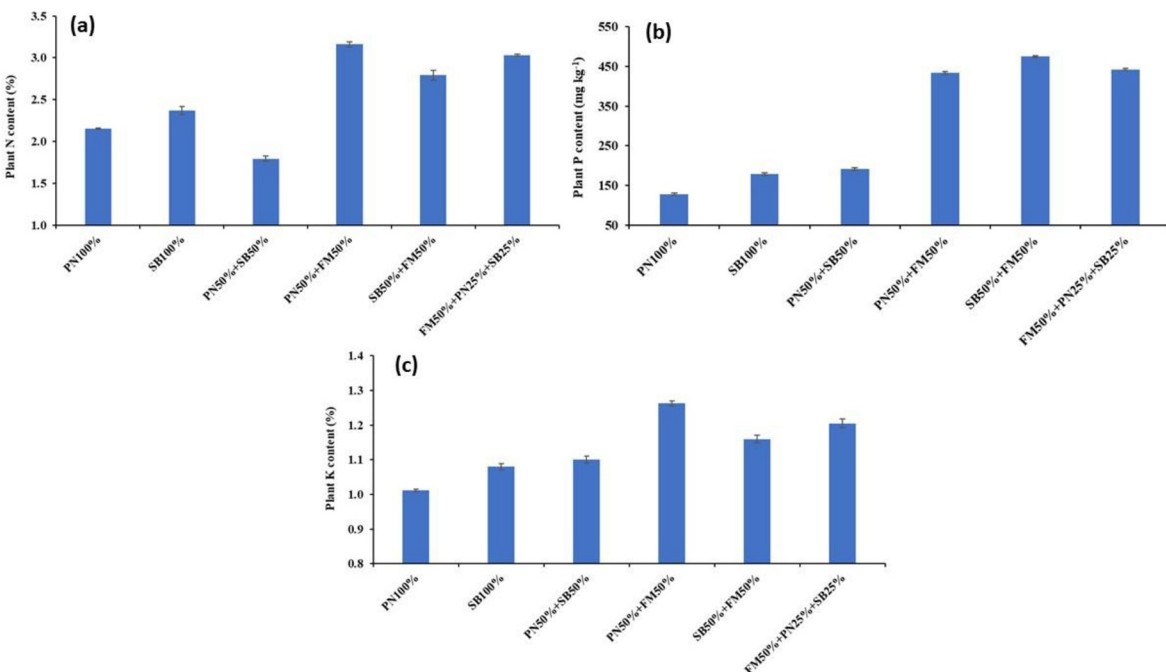

**Fig 2. Effect of substrate composition on spinach macronutrient content.** Error bars indicate standard errors from mean (n = 3). PN (pine needles); FM (farmyard manure); EM (effective microorganism).

The highest N % (1.54) P mg kg$^{-1}$ (0.064) and K % (0.819) were obtained at PN50% +FM50% while the lowest nutrient contents were recorded in SB100% media (Table 2). Adding FM to PN significantly increased macronutrient contents in media compared to the SB and PN alone or their combination. The macronutrient contents were significantly higher in SB50% +FM50% and PN25%+FM50%+SB25% than the rest of the treatment combinations (Table 2).

Micronutrient content in potting media has been shown in Table 3. The highest contents of Cu, Fe, and Mn were recorded for PN50%+FM50% media while the lowest values were obtained for SB100% (Table 3). In the case of Zn, the highest values were noted for SB50% +FM50% media, and the lowest Zn contents were recorded for PN100% media. Using PN or SB alone or in combination produced the lowest nutrient contents in potting media however, adding FM to SB and PN significantly increased micronutrient contents (Table 3).

**Table 1. Influence of substrate composition on spinach micronutrient content.**

| Treatments | Cu (mg kg$^{-1}$) | Zn (mg kg$^{-1}$) | Mn (mg kg$^{-1}$) | Fe (mg kg$^{-1}$) |
|---|---|---|---|---|
| PN100% | 3.19±0.05 d | 14.4±1.29 d | 22.9±0.92 d | 30.2±1.41 c |
| SB100% | 3.34±0.02 c | 23.4±0.54 c | 18.2±0.15 e | 22.7±0.96 d |
| PN50%+SB50% | 3.16±0.03 d | 16.2±0.95 d | 19.3±0.58 e | 26.6±0.44 c |
| PN50%+FM50% | 4.36±0.01 a | 42.5±0.87 b | 52.9±0.09 a | 81.2±1.33 a |
| SB50%+FM50% | 4.39±0.01 a | 48.3±0.77 a | 42.4±0.46 c | 73.7±1.67 b |
| FM50%+PN25%+SB25% | 4.21±0.02 b | 43.5±0.58 b | 44.9±0.59 b | 74.8±1.03 b |

Each value is the mean of three replicates ± SD (n = 3). Different letters after means show significant differences (Tukey's HSD test, p ≤ 0.05). PN: pine needles; FM: farmyard manure; EM: effective microorganism.

**Table 2. Effect of the substrate composition on chemical properties of potting mix.**

| Treatments | pH | EC (mSm⁻¹) | N (%) | P (%) | K (%) |
|---|---|---|---|---|---|
| PN100% | 6.12±0.05 b | 78.7±3.36 d | 0.62±0.03 c | 0.029±0.00 b | 0.498±0.01 b |
| SB100% | 6.13±0.03 b | 91.2±1.29 c | 0.44±0.04 d | 0.028±0.00 b | 0.235±0.05 c |
| PN50%+SB50% | 6.12±0.04 b | 87.8±1.10 c | 0.56±0.04 cd | 0.029±0.00 b | 0.308±0.01 c |
| PN50%+FM50% | 6.69±0.06 a | 114±2.72 b | 1.54±0.06 a | 0.064±0.00 a | 0.819±0.11 a |
| SB50%+FM50% | 6.80±0.10 a | 125±2.79 a | 1.26±0.05 b | 0.062±0.00 a | 0.646±0.08 ab |
| PN25%+FM50%+SB25% | 6.76±0.06 a | 117±2.71 b | 1.34±0.06 b | 0.063±0.01 a | 0.680±0.07 a |

Each value is the mean of three replicates ± SD (n = 3). Different letters after means show significant differences (Tukey's HSD test, p ≤ 0.05). PN: pine needles; FM: farmyard manure; EM: effective microorganism.

## 4. Discussion

Unbalanced nutrition is responsible for reducing the soilless culture [35], therefore, differences in the growth of plants are linked to the chemical and physical features, though, understanding of these features is still not clear. Characterization of substrate and their statistical analysis depicted that a combination of FM with PN and SB could be an alternative for the substitution of imported material (sphagnum peat moss and coconut coir) using the soilless harvest technique for vegetable production. During this experiment, spinach plants were grown in different substrates alone or combinations. All the combinations having FM, PN, and SB favored the plant growth, and a significant difference was recorded with the treatments having only PN, SB, or their combination which is justified by the characteristics of these wastes.

Although many characteristics of potting media influence plant growth, pH is considered among the most important factors because it is directly connected to nutrient availability [36]. The pH of the FM is generally slightly higher to grow vegetables however, the addition of an acidic material like PN with FM gives favorable pH values for soilless growing media. The current study displayed that, except for SB50%+FM50%, pH for other combinations was in the desired range (5.8–6.8) having all essential nutrients available and considered suitable for most crops [37]. Optimum levels of pH differ with plant species (Ingram & Henley, 1993). Lower pH can lead to ammonium toxicity while at too high pH, micronutrients become deficient [36]. Potting media with suitable pH ranges between 5.3–6.8 which generally allows better nutrient availability without the danger of toxicity and these results follow our results [38]. Important factors influencing pH are the substrate, quality of irrigation water, the pH of fertilizer solutions, and the plant species. The *P. roxburghii* needles are acidic in nature which is

**Table 3. Influence of substrate composition on micronutrient content in potting mix.**

| Treatments | Cu (mg kg⁻¹) | Fe (mg kg⁻¹) | Mn (mg kg⁻¹) | Zn (mg kg⁻¹) |
|---|---|---|---|---|
| PN100% | 5.03±0.6 c | 68.7±4.0 d | 16.8±1.4 cd | 22.6±1.2 c |
| SB100% | 3.88±0.3 c | 41.7±3.9 e | 12.9±1.7 d | 31.2±1.7 c |
| PN50%+SB50% | 4.56±0.3 c | 50.8±2.9 e | 14.7±1.2 d | 27.8±1.3 c |
| PN50%+FM50% | 10.4±1.0 a | 660±4.7 a | 28.2±2.6 a | 55.5±2.1 b |
| SB50%+FM50% | 8.15±0.8 b | 625±3.2 c | 22.2±2.4 bc | 66.9±2.5 a |
| PN25%+FM50%+SB25% | 9.00±1.0 ab | 642±2.7 b | 24.3±1.3 ab | 60.9±2.6 ab |

Each value is the mean of three replicates ± SD (n = 3). Different letters after means show significant differences (Tukey's HSD test, p ≤ 0.05). PN: pine needles; FM: farmyard manure; EM: effective microorganism.

why when these needles were used as potting media components pH remained lower as compared to other materials whereas, during the decomposition, different organic acids were produced which resultantly reduced the pH as observed in this study. Electrical conductivity depicts soluble salts in a medium and lower soluble salt contents are preferable for vegetable production [39] because higher EC inhibits biological activity, and plant growth [40]. In the current study, EC values were in the acceptable range (78.7–125 mScm$^{-1}$) for all the treatments. Substrates with higher EC such as FM mixed with compost up to 50%, did not pose any detrimental impact on plant growth [37]. Critical ranges for EC could be different in different countries for EC of potting media.

All the combinations of growing media in this study were suitable for vegetable production as nutrient contents were in the acceptable ranges. Lower macro- and micronutrient contents were observed in the treatments where SB was used in higher ratios (100%) which contains lower nutrient contents as compared to other materials used in combination (Tables 2, 3). Generally, a low ratio of FM in combination resulted in lower nutrient content in growing media. A combination of FM with PN and SB showed better results as compared to sole PN, SB, or their combination. It may be due to lower EC and pH coupled with higher C: N ratio in these substrates. Crop attributes and their nutrient contents were increased when FM was used in combination with other inorganic amendments [41]. This is possible because of pH and EC adjustments along with higher nutrient content provided by the FM.

Specifically, the treatments PN50%+FM50% and PN25%+FM50%+SB25% exhibited the highest performance. It could be linked with higher microbial activity and community composition, which enhances substrate mineralization which in turn increases the nutrient concentration [35]. Therefore, the interaction between substrates affects plant growth attributes (height, leaves per plant, shoot fresh- and dry weight) and chemical attributes (pH and EC). Hence, the important aspect of potting media is to maintain the proper quantity of PN and SB in the mixture. By potting media, essential nutrients adherence to PN and SB can decrease the leaching of nutrients, favoring plant growth [42]. Furthermore, the addition of SB increased essential nutrient contents (N, P, and K) in the soil [43]. According to previous research, smaller quantities of SB in the substrates positively affected all the studied parameters [44] and similar results were obtained during the current study, though there was inconsistency between the SB ratio and plant growth [45]. Sugarcane bagasse enhances the physico-chemical attributes of potting media which promote plant growth [46].

The results of the current study are consistent with available literature which reports that FM contains higher nutrient contents and retains water during plant growth. Addition of SB and PN in the desired amount also favors plant performance. Sugarcane bagasse is commonly used in soilless culture however, in this study, it was added directly to assess its impact and observed positive impacts on plant growth as reported by [47]. During the experiments, it was observed that substrate composition has an impact on substrate pH. like the addition of PN produced acidic values but it remained within the desired range in all combinations. The versatility of PN and SB has also acted as organic matter sources and nutrient contents (macro- and micro) in combination with FM. Pine needles and SB in combination with FM improve the physico-chemical and biological characteristics of media [46].

Higher growth attributes and yield of plants grown in soilless potting media were also reported [48]. Similar results were reported by [49] that potting media having different organic materials produced higher plant biomass compared to potting media having a single ingredient and decreased the detrimental impact of heavy metals [50]. Growth parameters of plants vary owing to the physicochemical properties of substrate characteristics [10]. During this study, better spinach growth under greenhouse conditions was obtained with treatments PN50%+FM50% followed by PN25%+FM50%+SB25%. The increasing population contributes

to the mismanagement of organic waste, necessitating proper handling. The study's primary goal was to identify viable organic waste for growing vegetables that can reduce the environmental effect by producing value-added products.

## 5. Conclusion

During this study, various organic wastes, namely PN, FM, and SB, were examined to determine the optimal ratio for a soilless potting medium. Analyses revealed that among these agro-industrial wastes, FM exhibited a higher quantity of nutrient contents compared to PN and SB. The chemical composition of the raw materials affirmed that an effective substrate should consist of PN50%+FM50% and PN25%+FM50%+SB25% for growing spinach. The study showcased that altering the composition of the substrate could effectively regulate spinach plant growth in a soilless culture. Notably, the PN50%+FM50% treatment demonstrated the highest concentrations of both macro- (N, P, and K) and micronutrients (Fe, Cu, Mn, and Zn) in the potting media post-harvest. The ANOVA analysis revealed distinct differences among the substrates for various growth traits and chemical properties. Samples with the most favorable performance were those containing PN50%+FM50%, followed by PN25%+FM50% +SB25%. The results suggested that these agro-industrial wastes could contribute to cost reduction in soilless farming. However, further studies are recommended to explore these substrates concerning their particle size and their effectiveness in growing media.

## Acknowledgments

The authors express their gratitude to the Institute of Soil & Environmental Science, PMAS-Arid Agriculture University, Rawalpindi for providing laboratory and greenhouse for analysis and experiments.

## Author Contributions

**Conceptualization:** Tajwar Alam, Arshad Nawaz Chaudhry, Chaudhry Muhammad Subhan, Ghulam Jilani.

**Investigation:** Tajwar Alam.

**Writing – original draft:** Tajwar Alam, Ghulam Jilani.

**Writing – review & editing:** Muhammad Ikram, Arshad Nawaz Chaudhry, Chaudhry Muhammad Subhan, Khaled D. Alotaibi, Zia-Ul -Haq, Muhammad Sohail Yousaf, Hasan Pervej Ahmed, Nida Fatima, Muhammad Shakir Farooq, Mohammad Naseem, Muhammad Ishaq.

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
