## [Decision Letter · Decision Letter 0]

22 Feb 2024

PONE-D-24-02845Utilization of organic-residues as potting media: Physico-chemical characteristics and their influence on vegetable productionPLOS ONE

Dear Dr. Alam,

Thank you for submitting your manuscript to PLOS ONE. After careful consideration, we feel that it has merit but does not fully meet PLOS ONE’s publication criteria as it currently stands. Therefore, we invite you to submit a revised version of the manuscript that addresses the points raised during the review process.

We look forward to receiving your revised manuscript.

Kind regards,

Taimoor Hassan Farooq

Academic Editor

PLOS ONE

Journal Requirements:

2. In this instance it seems there may be acceptable restrictions in place that prevent the public sharing of your minimal data. However, in line with our goal of ensuring long-term data availability to all interested researchers, PLOS’ Data Policy states that authors cannot be the sole named individuals responsible for ensuring data access (http://journals.plos.org/plosone/s/data-availability#loc-acceptable-data-sharing-methods).

3. Please amend the manuscript submission data (via Edit Submission) to include author "Zia-ul-Haq". 

**Additional Editor Comments:**

We invite you to submit a revised version of the manuscript that addresses the points raised during the review process.

Reviewers' comments:

Reviewer's Responses to Questions

**Comments to the Author**

1. Is the manuscript technically sound, and do the data support the conclusions?

Reviewer #1: Yes

Reviewer #2: No

2. Has the statistical analysis been performed appropriately and rigorously? 

Reviewer #1: Yes

Reviewer #2: Yes

3. Have the authors made all data underlying the findings in their manuscript fully available?

Reviewer #1: Yes

Reviewer #2: Yes

4. Is the manuscript presented in an intelligible fashion and written in standard English?

Reviewer #1: Yes

Reviewer #2: Yes

5. Review Comments to the Author

Reviewer #1: The article reports an innovative approach to utilize organic waste (pine needles, sugarcane bagasse, and farmyard manure) for making soilless potting media, which looks useful for organic food production. Basic analytical work and greenhouse experiments were undertaken precisely and reported with detailed methodology. Study results have been discussed to conclude that the technology bears commercial value. This manuscript merits publication in PLOS ONE after major improvements as suggested below:

1. I suggest the author remove abbreviated words from the abstract and revise it. Kindly use full names here.

2. Line 37-40: Kindly revise it. I found some inconsistencies in these sentences.

3. I suggest the author revise the keywords and it should be not the same as the title of the manuscript.

4. Line 81: Do not start the sentence with the reference. It looks awkward and not according to scientific standards.

5. At least mention three objectives and hypotheses of your current research and correlate them with the discussion section.

6. Check the references in the text and the reference section. Make sure it should be according to the journal format.

7. I suggest you some new references related to your study. Kindly cite them in your research article which will increase the worth of your article.

Microplastics meet invasive plants: Unraveling the ecological hazards to agroecosystems

Impacts of soil microplastics on the crops: A review. Applied soil ecology

Harnessing soil carbon sequestration to address climate change challenges in agriculture

Dynamic changes of soil nematodes between bulk and rhizosphere soils in the maize (Zea mays L.)/alfalfa (Medicago sativa L.) intercropping system

The detrimental effects of heavy metals on tributaries exert pressure on water quality, Crossocheilus aplocheilus, and the well-being of human health

Influence of soil microplastic contamination and cadmium toxicity on the growth, physiology, and root growth traits of Triticum aestivum L.

8. The conclusion statement should be comprehensive. Kindly revise it.

9. The introduction looks lengthy and some of the statements are repeated, which should be avoided.

10. Also, include the detailed procedure for Total N determination.

Reviewer #2: MS Title: Utilization of organic-residues as potting media: Physico-chemical characteristics and their influence on vegetable production

General comments

This manuscript addresses an important issue by processing the waste pine needles and sugarcane bagasse along with farmyard manure to develop soilless potting media. Experiments are conducted appropriately, However, The manuscript lacks explicit presentation of control data for each experiment conducted by the authors. It is imperative to include control conditions where no organic matter was added to the soil to facilitate meaningful comparisons across different treatments. In order for the manuscript to undergo thorough review, it is essential that control data be provided for comparison. Without such data, meaningful evaluation of the experiments conducted would be compromised.

6. PLOS authors have the option to publish the peer review history of their article (what does this mean?). If published, this will include your full peer review and any attached files.

Reviewer #1: **Yes: **Babar Iqbal

Reviewer #2: No

---

## [Author Response · Author response to Decision Letter 0]

13 Mar 2024

Response to Reviewers’ Comments

Reference: PONE-D-24-02845

Title: Utilization of organic-residues as potting media: Physico-chemical characteristics and their influence on vegetable production

NOTE: Revised/added text as per review comments has been highlighted in the revised manuscript through track changes.

Authors acknowledge the comments by respectable reviewers and editor to improve the quality of the manuscript. Thank you very much for these efforts and for sparing valuable time.

Reviewer #1: 

Query 1. I suggest the author remove abbreviated words from the abstract and revise it. Kindly use full names here.

Response: The abbreviation from the abstract has been removed and revised with full names as suggested.

Query 2. Line 37-40: Kindly revise it. I found some inconsistencies in these sentences.

Response: Sentences have been revised. 

Query 3. I suggest the author revise the keywords and it should be not the same as the title of the manuscript.

Response: Needful has been done.

Query 4. Line 81: Do not start the sentence with the reference. It looks awkward and not according to scientific standards.

Response: Suggestion incorporated.

Query 5. At least mention three objectives and hypotheses of your current research and correlate them with the discussion section.

Response: Objectives and hypotheses of research have been included in the last paragraph of the introduction.

Query 6. Check the references in the text and the reference section. Make sure it should be according to the journal format.

Response: Needful has been done.

Query 7. I suggest you some new references related to your study. Kindly cite them in your research article which will increase the worth of your article.

Response: Suggestion incorporated.

Query 8. The conclusion statement should be comprehensive. Kindly revise it.

Response: Conclusion statement has been revised as per suggestion.

Query 9. The introduction looks lengthy and some of the statements are repeated, which should be avoided.

Response: Needful has been done throughout the introduction as per suggestion.

Query 10. Also, include the detailed procedure for Total N determination.

Response: A detailed procedure of total N determination has been added under the heading of chemical analysis in detail.

Reviewer #2: 

Query 1. The manuscript lacks explicit presentation of control data for each experiment conducted by the authors. It is imperative to include control conditions where no organic matter was added to the soil to facilitate meaningful comparisons across different treatments. In order for the manuscript to undergo thorough review, it is essential that control data be provided for comparison. Without such data, meaningful evaluation of the experiments conducted would be compromised.

Response: We appreciate your insightful comments. Since our experimental setup was soilless, it may not be appropriate to include a soil control. Therefore, we compared the available organic residues (FM, PN, and SB) in all possible combinations to fulfill the requirement for a baseline comparison. Thank you for highlighting the significance of control data. We are dedicated to offering pertinent and enlightening data to support a comprehensive assessment of the paper.

---

## [Decision Letter · Decision Letter 1]

20 Mar 2024

PONE-D-24-02845R1Utilization of organic-residues as potting media: Physico-chemical characteristics and their influence on vegetable productionPLOS ONE

Dear Dr. Alam, Thank you for submitting your manuscript to PLOS ONE. After careful consideration, we feel that it has merit but does not fully meet PLOS ONE’s publication criteria as it currently stands. Therefore, we invite you to submit a revised version of the manuscript that addresses the points raised during the review process.

We look forward to receiving your revised manuscript.

Kind regards,

Taimoor Hassan Farooq

Academic Editor

PLOS ONE

Journal Requirements:

**Additional Editor Comments:**

Please focus on reviewer comments about figures.

Reviewers' comments:

Reviewer's Responses to Questions

**Comments to the Author**

1. If the authors have adequately addressed your comments raised in a previous round of review and you feel that this manuscript is now acceptable for publication, you may indicate that here to bypass the “Comments to the Author” section, enter your conflict of interest statement in the “Confidential to Editor” section, and submit your "Accept" recommendation.

Reviewer #1: All comments have been addressed

Reviewer #3: (No Response)

2. Is the manuscript technically sound, and do the data support the conclusions?

Reviewer #1: Yes

Reviewer #3: Yes

3. Has the statistical analysis been performed appropriately and rigorously? 

Reviewer #1: Yes

Reviewer #3: Yes

4. Have the authors made all data underlying the findings in their manuscript fully available?

Reviewer #1: Yes

Reviewer #3: Yes

5. Is the manuscript presented in an intelligible fashion and written in standard English?

Reviewer #1: Yes

Reviewer #3: Yes

6. Review Comments to the Author

Reviewer #1: (No Response)

Reviewer #3: Your efforts in refining the manuscript have been duly recognized.

I suggest you consider incorporating more figures than tables into your manuscript. Figures not only enhance the visual appeal of your paper but also provide a succinct and informative way to present your findings. I suggest labeling the figures as "Fig. 1a, 1b, etc. for each type of parametrs" for clarity and consistency throughout the document. Additionally, I encourage you to include a study area figure in your paper. This section will provide essential context for your research, helping readers better understand the geographical or thematic scope of your study.

7. PLOS authors have the option to publish the peer review history of their article (what does this mean?). If published, this will include your full peer review and any attached files.

Reviewer #1: No

Reviewer #3: No

---

## [Author Response · Author response to Decision Letter 1]

22 Mar 2024

Response to Reviewers’ Comments

Reference: PONE-D-24-02845R1

Title: Utilization of organic-residues as potting media: Physico-chemical characteristics and their influence on vegetable production

NOTE: Revised text as per review comments has been highlighted in the revised manuscript through track changes.

The authors sincerely appreciate the insightful comments provided by the esteemed editor and reviewers, which have greatly contributed to enhancing the quality of the manuscript. We are grateful for their valuable time and efforts dedicated to the review process. Thank you.

Reviewer #3: 

Query. I suggest you consider incorporating more figures than tables into your manuscript. Figures not only enhance the visual appeal of your paper but also provide a succinct and informative way to present your findings. Additionally, I encourage you to include a study area figure in your paper. 

Response: As per the suggestion of the respected reviewer Table 1 titled “Effect of substrate composition on spinach growth and yield attributes” (Plant height (cm), Fresh biomass (g plant−1), Dry biomass (g plant−1), Leaves plant−1) has been converted into figures to enhance the visual appeal. Moreover, the study area information has been provided in the materials and methods section (2.1. Site description), and the study area figure has been provided in Fig 1.

---

## [Editor Report · Decision Letter 2]

28 Mar 2024

Utilization of organic-residues as potting media: Physico-chemical characteristics and their influence on vegetable production

PONE-D-24-02845R2

Dear authors,

We’re pleased to inform you that your manuscript has been judged scientifically suitable for publication and will be formally accepted for publication once it meets all outstanding technical requirements.

Kind regards,

Taimoor Hassan Farooq

Academic Editor

PLOS ONE

Additional Editor Comments (optional):

Dear authors,

I have gone through your manuscript in response to the reviewers' comments. After carefully reviewing the revisions made based on the reviewers' comments, I am pleased to say that your paper has met the standards and requirements for publication. The revisions addressed the concerns raised by the reviewers effectively, enhancing the clarity, coherence, and overall quality of the manuscript.

I am writing to inform you that your manuscript has been accepted for publication.
---

## [Editor Report · Acceptance letter]

21 May 2024

PONE-D-24-02845R2 

PLOS ONE

Dear Dr. Alam, 

I'm pleased to inform you that your manuscript has been deemed suitable for publication in PLOS ONE. Congratulations! Your manuscript is now being handed over to our production team.

Kind regards, 

on behalf of

Taimoor Hassan Farooq 

Academic Editor

PLOS ONE